# Cycles of Andean mountain building archived in the Amazon Fan

Cody C. Mason [1] ✉, Brian W. Romans [2], Molly O. Patterson[3], Daniel F. Stockli[4] & Andrea Fildani[5]

Cordilleran orogenic systems have complex, polycyclic magmatic and deformation histories, and the timescales and mechanisms of episodic orogenesis are still debated. Here, we show that detrital zircons (DZs) in terrigenous sediment from the late Pleistocene Amazon Fan, found at the terminus of the continent-scale Amazon River-fan system, record multiple, distinct modes of U-Pb crystallization ages and U-Th/He (ZHe) cooling ages that correlate to known South American magmatic and tectonic events. The youngest ZHe ages delineate two recent phases of Andean orogenesis; one in the Late Cretaceous – Paleogene, and another in the Miocene. Frequency analyses of the deep-time Phanerozoic record of DZ U-Pb and ZHe ages demonstrate a strong 72 Myr period in magmatic events, and 92 Myr and 57 Myr periods in crustal cooling. We interpret these results as evidence of changes in upper and lower plate coupling, associated with multiple episodes of magmatism and crustal deformation along the subduction-dominated western margin of South America.

Globally, cordilleran systems with volcanic arcs exhibit episodic cycles of magmatism, which are often constrained by the measurement of U-Pb ages of zircons in igneous and detrital material (Paterson and Ducea[1]). Conceptual models link episodic magmatism in cordilleran arc systems to a suite of related processes in the mantle and upper plate, including tectonic activity in the orogen, and resulting surface processes (DeCelles et al.[2]). For example, one version of the orogenic cycle consists of periods of high magmatic flux associated with underthrusting of melt-fertile lower crust, delamination of a dense mafic orogenic root from the upper plate, and subsequent rapid isostatic uplift, tectonic growth of the fold-thrust belt, and increased orogen relief associated with enhanced erosion rates (DeCelles et al.[2,3]). Datasets of multiple thermochronometers from clastic detritus have previously been used to propose anti-correlation between the timing of magmatic flux events, and tectonic denudation of an orogen (Carrapa and DeCelles[4]). And while orogenic cycles are proposed to occur at quasi-periodic timescales of ~25–50 Myr (DeCelles et al.[2,3]),

or ~70 Myr (Sundell et al.[5]), the nature and timing of erosional response to orogenic cycles are still unclear.

Thermal events in the orogenic belt are preserved in the detritus shed from cordilleran systems into their associated basins (Reiners and Brandon[6]), and, in some cases, ultimately transferred to passive margin marine stratigraphic archives (Mason et al.[7]; Fildani et al.[8]). Deep-sea fans—large accumulations dominated by terrigenous sediment—record the physical and chemical signatures of their linked continent-scale river catchments (Hessler and Fildani[9]). The Amazon Fan, offshore NE South America (Fig. 1), contains sediments that originated throughout its catchment, with a predominance of detritus sourced from the central and northern Andes (McDaniel et al.[10]; Mason et al.[7]). Since this terrigenous detritus records the tectonic history of the catchment, it is useful in the recovery of past thermal and deformation events in the cordilleran system. The application of low-temperature detrital thermochronometry could help clarify the tempo of magmatic and thermal events experienced by South America's active western margin over numerous timescales.

[1]Department of Natural Sciences, University of West Georgia, Carrollton, GA, USA. [2]Department of Geosciences, Virginia Tech, Blacksburg, VA, USA. [3]Department of Geology, Binghamton University, State University of New York, Binghamton, NY, USA. [4]Jackson School of Geosciences, University of Texas, Austin, TX, USA. [5]The Deep Time Institute, Danville, CA, USA. ✉e-mail: cmason@westga.edu

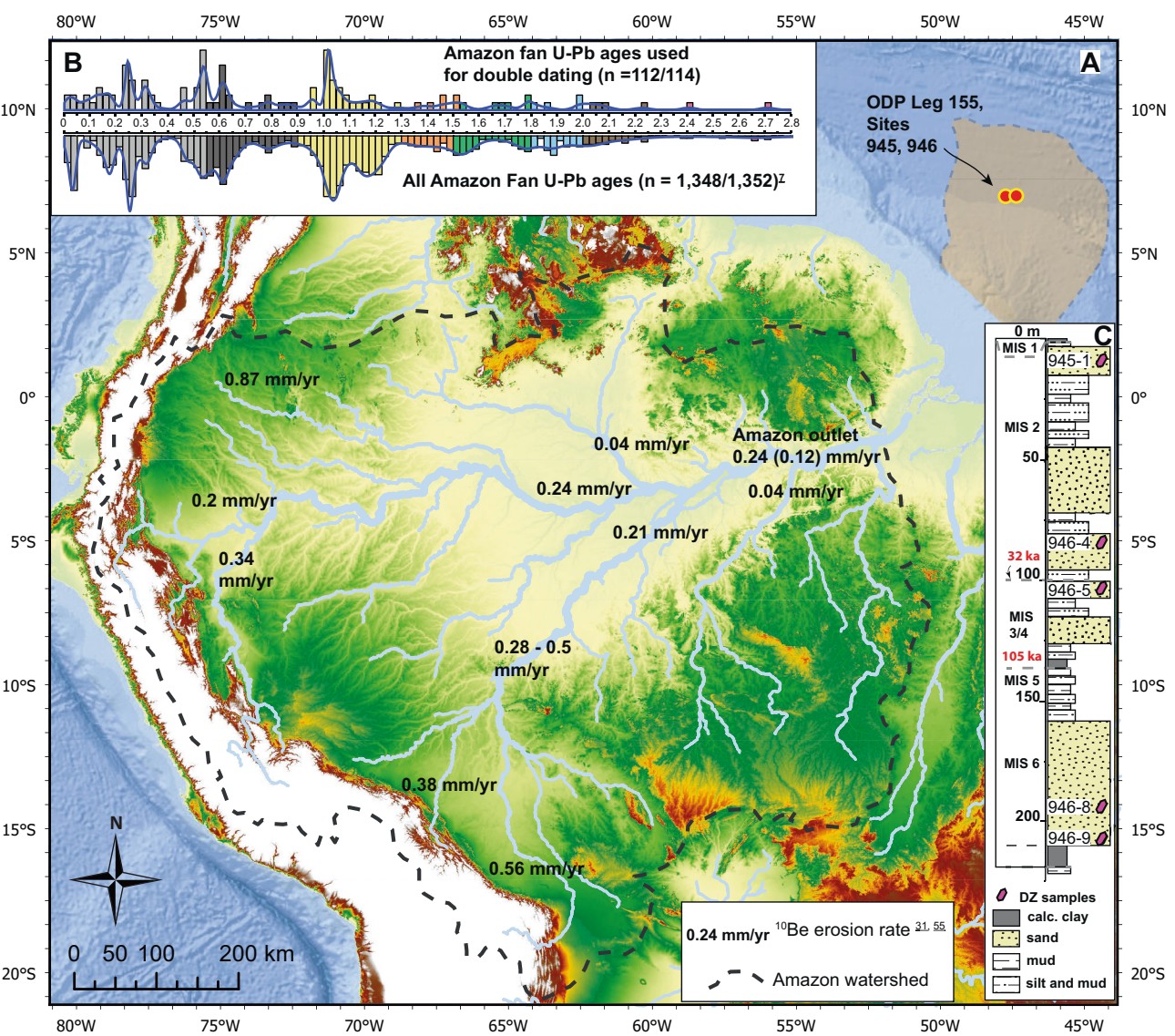

**Fig. 1 | The continent-scale Amazon River to fan system. A** Elevation map of the Amazon drainage basin and approximate extents of the Amazon deep-sea fan. Tributaries and main-stem Amazon River presented with [10]Be-derived erosion rates, and estimates of meteoric [10/9]Be-derived erosion rates in parenthesis (Wittmann et al.[31,55] respectively). **B** Detrital zircon U-Pb ages from the Pleistocene Amazon fan used in this study, from Mason et al.[7]. **C** Composite lithostratigraphy from Ocean Drilling Program (ODP) Leg 155, sites 945 and 946 with schematic detrital zircon sample locations (after Mason et al.[7]).

Here, we characterize the thermotectonic histories experienced by detrital zircons (DZs) found in the late Pleistocene Amazon Fan through the application of combined DZ U-Pb and U-Th/He double dating (U-Pb/ZHe double dating; $n = 114$ individual DZs). We perform frequency analyses of U-Pb and ZHe age data, and present the detrital geochronologic and frequency analyses results in the context of: (1) provenance interpretations of Amazon Fan sediment, and (2) past orogenic events of the South American continent, including generation of zircons during phases of magmatism, metamorphism, and their low-temperature cooling histories associated with the exhumation of the Andes and predecessor orogenic systems.

Western South America is dominated by the high-elevation, cordilleran system formed through the subduction of oceanic lithosphere, arc volcanism, and deformation of the overlying South American plate. The orogenic system is divided, from west to east, into a forearc region consisting of a western cordillera containing the magmatic arc, and a retroarc region, consisting of an eastern cordillera, in the case of the central Andes a broad high-elevation plateau, and an active retroarc fold thrust belt (Horton[11]).

The recent phase of Andean orogenesis initiated due to subduction of the Nazca plate beneath the South America plate during the late Mesozoic, and may have progressed in time from the northern Andes toward the central and southern Andes (Chen et al.[12]). During the Paleogene, rates of shortening in the central Andes increased dramatically (Armijo et al.[13]; and references therein; Horton[14]), as did the elevation of the central Andes (McQuarrie et al.[15]) culminating in the Miocene with rise of the central Andean Plateau (Garzione et al.[16]; Sundell et al.[17]). Andean retroarc foreland basin fill records increased accommodation and initiation of widespread coarse grained sedimentation starting in the Late Cretaceous to early Paleogene, and is interpreted to be the signal of shortening induced lithospheric flexure (Horton[11] and references therein). A geographically widespread hiatus or cessation of coarse grained sedimentation occurred across the Eocene, with renewed coarse grained sedimentation recorded in retroarc foreland basins starting in the Oligocene (Horton[11])

Subduction related deformation along the pre-Andean margin was polycyclic, long lived and regional in extent (Chew[18]). Early orogenic events experienced by western South America are the

Neoproterozoic through Paleozoic assembly of Gondwana, including the Pan African-Brasiliano (Neoproterozoic), the Pampean (~Cambrian) and the Famatinian (~Ordovician) orogenies, respectively (Ramos et al.[19]; Chew et al.[18]). These widespread events were followed by phases of Permian-Triassic metamorphism (Chew et al.[18]), and Jurassic magmatism along the western margin of South America (Kontak et al.[20]). During the Mesozoic, prior to widespread Andean shortening, extension and post-rift thermal subsidence were the dominant modes of lithospheric deformation along much of what is now the central Andes (Horton[14]), leading to the formation of large sedimentary basins along strike in the Cretaceous (Horton[11]).

Andean streams are tributaries to the Amazon River, which transfers clastic detritus from a wide array of lithologies, representing a large span of geologic time, depositional environments, and igneous and metamorphic facies (Chew et al.[21]). Northern Andean streams with high elevation headwaters in southern Colombia and Ecuador tap into an active volcanic arc, Paleogene and Cretaceous sedimentary units, Neoproterozoic metamorphic units, Triassic metamorphic, and Jurassic volcanic and plutonic units in the retro arc fold-thrust belt (Jackson et al.[22]; Gomez Tapias et al.[23]).

A magmatic gap related to Peruvian flat slab subduction exists between ~2° and 15° S (Bernal et al.[24]; Gomez Tapias et al.[23]). There, Andean streams in Peru tap Mesozoic and Paleozoic sedimentary, volcanic, and metamorphic units in the eastern cordillera, and Cenozoic, Cretaceous, and Jurassic sedimentary units in the fold-thrust belt (Gomez Tapias et al.[23]). South of ~15° S, in the western cordillera of Peru, lies an active volcanic arc. However, most volcanic detritus is transferred to internally drained, high elevation basins within the Altiplano of Peru and Bolivia. The eastern cordillera, and fold-thrust belt of southern Peru and Bolivia are dominated by Cenozoic, late Mesozoic, and extensive Paleozoic sedimentary units (Gomez Tapias et al.[23]).

The modern east-flowing Amazon River is a relatively recent feature. An ancestral Amazon River once flowed east to west, supplying sediment from the craton to Andean retroarc foreland basins, until a phased reversal was completed in the Miocene, as marked by the transfer of Andean detritus to the Atlantic Ocean (Hoorn et al.[25]). Consequently, DZ age spectra of sediment from modern rivers of the Amazon system display evidence for recycling of zircons originally sourced from the craton (Pepper et al.[26]; Mason et al.[7]; Capaldi et al.[27]; Jackson et al.[22]). Specifically, modern river sediments from the Andes may contain DZs with age components related to the Meso-Neoproterozoic Sunsás orogen (Grenville equivalent), the Neoproterozoic to Cambro-Ordovician Famatinian–Pampean orogens, and U-Pb ages related to periods of Mesozoic through Cenozoic magmatic arc activity along western South America (Horton et al.[28]; Capaldi et al.[27]). DZs with U-Pb ages >1300 Ma correspond to cratonic terranes and are documented in modern Andean sourced rivers and in bedrock sources within the northern to central Andes (George et al.[29]). Pampean aged DZs are sometimes referred to as Pan African–Brasiliano, an orogeny that affected eastern South America, and which can also be found in modern Andean streams. Due to long term sediment recycling in the Andes and sedimentary basins of the fold-thrust belt, the provenance of DZs in the lower Amazon River and Fan becomes difficult to discern (Mason et al.[7]). We address this issue with the application of double dating of DZs, where recent tectonic exhumation may be revealed by the low-temperature cooling history of cratonic aged DZs sourced from the Andes.

Onshore, catchments within the Amazon Basin draining high relief mountainous areas represent a small fraction of the total Amazon drainage basin area (~12%; Gibbs[30]), yet contribute a disproportionate amount of sediment to the Amazon fluvial system and related deep-sea fan (Fig. 1). Cosmogenic nuclide concentrations of river sediment show that Andean sourced rivers have erosion rates an order of magnitude higher than Amazon lowland rivers, over multi-millennial timescales

(See Fig. 1; Wittman et al.[31]). Direct observations of river suspended sediment loads (Meade et al.[32]), mud geochemistry from the Amazon Fan (McDaniel et al.[10]), and DZ provenance estimates along the lower Amazon River and deep-sea Fan suggest that a large proportion of sediment (~80–90%) may be sourced from the Andes (Mason et al.[7]). Differential zircon fertility of source terranes has been shown to influence DZ age distributions in some cases (Moecher and Samson[33]), which may interfere with provenance interpretations. However, denudation rates of Andean streams that are an order of magnitude higher than low-lying Amazonian streams are likely a dominant process driving provenance signatures in the Amazon Fan. Given the source of sediment in the Amazon Fan is largely from the cordilleran system, the Amazon Fan sedimentary record should preserve the long thermal history of the tectonically active western margin of South America.

## Results

### Detrital zircon U-Pb and U-Th/He double dating
U-Pb and ZHe ages from 114 DZs recovered from the Amazon Fan characterize the thermotectonic histories experienced by DZs in the Amazon system. Complete isotopic measurements may be found in Supplementary Datafile S1. Figure 2 displays a cross plot of ZHe age vs. U-Pb age, with associated kernel density estimates (KDEs) and histograms. Figure 3A–D displays cross plots of select U-Pb age groups (0–200 Ma, 200–400 Ma, 400–900 Ma, and 900–1300 Ma). Only two individual DZs resulted in ZHe ages greater than their associated U-Pb ages (Fig. 2), and are thus excluded from further analyses. In the following paragraphs, we report the resultant proportions of ZHe cooling ages, and relate these ZHe ages to their associated U-Pb age populations.

### First cycle volcanic zircons
Six first cycle volcanic DZs were identified by their overlapping U-Pb and ZHe ages (within analytical error; Campbell et al.[34]; Saylor et al.[35]). These first cycle volcanic DZs range in age from ~30 Ma to 585 Ma, with the majority (four of six) from Mesozoic and Cenozoic U-Pb age groups (Fig. 3A). Of the five DZs with Cenozoic U-Pb ages, only two have errors that overlap, and are first cycle volcanic zircons (*sensu stricto*). The remaining three Cenozoic DZs have ZHe ages that are several Myr younger (~3–10 Myr) than their associated U-Pb ages.

### U-Th/He age groups
Of the 112 accepted ZHe ages, 38% ($n = 42/112$) are Cenozoic, 22% ($n = 25/112$) are Mesozoic, 29% ($n = 32/112$) are Paleozoic, and 12% ($n = 13/112$) are Precambrian in age (Fig. 2). ZHe age density increase dramatically across the Ediacaran (<~600 Ma) and through the Phanerozoic, while very few DZs record cooling older than the beginning of the Pan African-Brasiliano orogeny (>~900 Ma). We document important ZHe age modes present in the dataset, from old to young (see also Fig. 2): (1) an Ediacaran mode (~575–600 Ma), (2) a distributed late Cambrian through early Carboniferous mode (~500–350 Ma), (3) a latest Carboniferous to middle Permian mode (~300–275 Ma), (4) a late Triassic to Early Jurassic mode (~225–175 Ma), (5) a minor Early Cretaceous mode (~110–125), (6) a Late Cretaceous through Paleocene mode (~75–50 Ma), and finally (7) a Miocene through recent mode (25–0 Ma). The most prominent population of ZHe ages falls in the middle – late Miocene, with a peak mode at ~12 Ma (Fig. 2).

### U-Th/He ages in context of U-Pb age groups
Relationships between DZ U-Pb and ZHe ages yield insights into the functioning of sediment routing systems and thermal histories experienced by their sediment loads. In the Amazon Fan, the majority of DZs with U-Pb ages >1300 Ma, often interpreted as cratonic DZs, display distributed ZHe cooling ages across the Late Neoproterozoic, Paleozoic, and middle Mesozoic (Fig. 2). However, several early – late

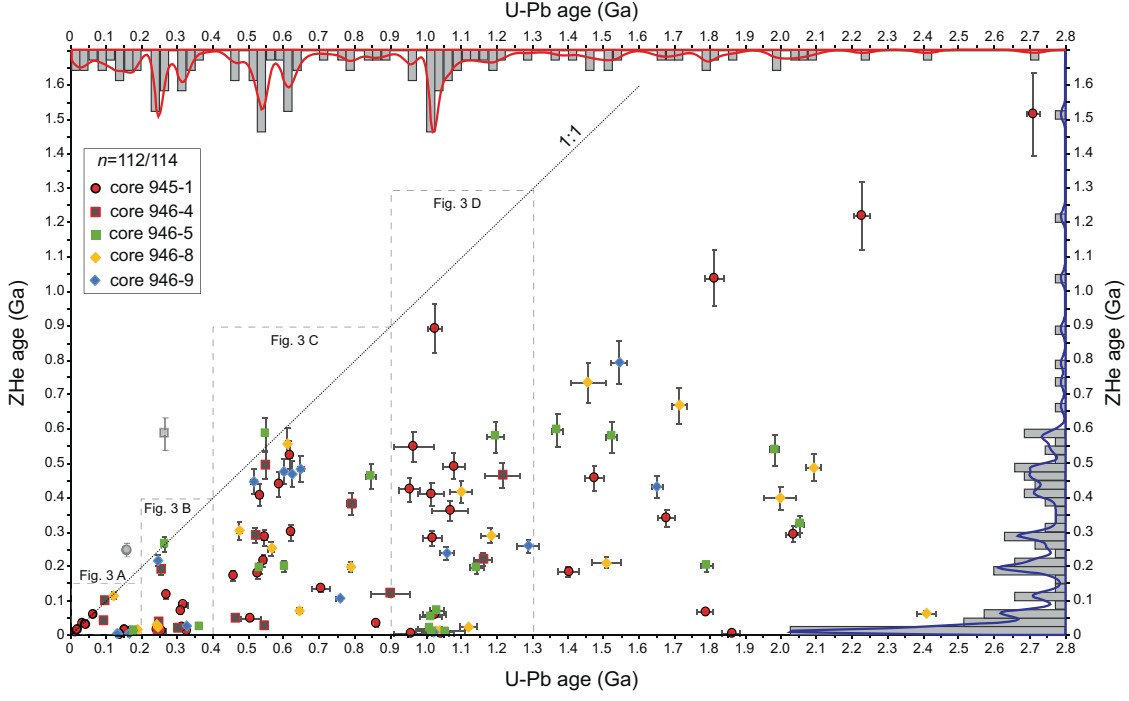

**Fig. 2 | Cross plot and kernel density estimates of U-Pb and U-Th/He double dates from detrital zircons recover from the late Pleistocene Amazon Fan (Leg 155, Sites 945, 946).** Age bins at 25 Ma increments. See Supplementary Fig. S1 for individual sample data from ODP cores 945 and 946. Error bars are two standard deviations.

Cenozoic ZHe ages occur in this oldest U-Pb age group, and record recent cooling for at least some proportion of age groups normally interpreted as craton derived.

A large proportion of DZs present in the Amazon Fan have U-Pb ages that correspond to the Sunsás orogeny (~29%; Mason et al.[7]; this study 24%; $n = 27/112$), yet only two ZHe ages record cooling associated with that period of time (900–1300 Ma). Instead, the Sunsás U-Pb age group is associated with ZHe cooling ages between ~200 and 600 Ma, and a distinct mode of Late Cretaceous through Neogene ZHe ages, ($n = 12/27$; ~5 –72 Ma; Fig. 3D), with the majority of youngest ZHe ages recording cooling during the late Miocene.

DZs with U-Pb ages between 400 and 900 Ma (28% of DZs in this study; Fig. 3C) are typically attributed to terranes that record orogenesis during the Pan African – Brasiliano orogenies (550–900 Ma), and the Cambro-Orovician Pampean – Famatinian orogenies. This U-Pb age group contains ZHe age modes at ~400–550 Ma, ~250–300 Ma, ~200 Ma, and a diffuse population of Early Cretaceous through Paleogene ZHe ages, but lacks Neogene ZHe ages.

DZs with U-Pb ages between 200 and 400 Ma (15% of DZs in this study; Fig. 3B) are likely associated with magmatism during Gondwanan orogenesis across the Carboniferous through Permian, with magmatism of equivalent age to Permo-Triassic Choiyoi volcanism, and with arc magmatism along the SW margin of Gondwana from the late Triassic through the early Jurassic. Devonian and Carboniferous U-Pb ages are associated with Cenozoic and Late Cretaceous ZHe ages, respectively (Fig. 3B). Permo-Triassic U-Pb ages are associated with one first cycle volcanic zircon, sparse late Triassic to early Cretaceous ZHe ages, and a high proportion of Cenozoic ZHe cooling ages. Overall, this U-Pb age group is dominated by Cenozoic ZHe ages.

Those DZs with U-Pb ages between 0 and 200 Ma (Fig. 3A) have a significant component of first cycle volcanic zircons, or are associated largely with late Cenozoic cooling ages (<50 Ma). Jurassic and early Cretaceous U-Pb ages are associated with one first cycle volcanic, and many Neogene ZHe ages. DZs with Late Cretaceous or younger U-Pb ages are first cycle volcanic, or associated with Paleogene through Neogene ZHe ages.

### Synthesis of detrital age data from the Amazon Fan

DZs sampled from the Amazon Fan integrate sedimentation across >150 kyr of the late Pleistocene glacial times, and taps into erosional sources across the Amazon Basin, with most sediment coming from the high elevation Andean streams. Through the relationships of U-Pb provenance and ZHe ages, we can begin to understand the thermal histories of distinct DZ source terranes.

Late Mesoproterozoic through Cambro-Ordovician U-Pb DZ age groups display several ZHe age modes that broadly record cooling during the early Paleozoic, the Permian, the Late Triassic – Early Jurassic, and the Late Cretaceous – Neogene (Fig. 3). The Proterozoic (900–1300 Ma) Sunsás U-Pb age group, much like the 400–900 Ma U-Pb age group, broadly records cooling across the Phanerozoic, but has an additional component of Neogene ZHe ages.

The 200–400 Ma U-Pb age group records cooling events in the Late Triassic – Early Jurassic, the Cretaceous, and the Cenozoic. The youngest U-Pb age group, 0–200 Ma, contains Jurassic and Cretaceous DZs that cooled in the Neogene, in the Cretaceous, and several Cenozoic first-cycle volcanics. Jurassic and Cretaceous DZs share a history of recent cooling with each of the other U-Pb age groups.

Collectively, DZs from the Amazon Fan record cycles of magmatism and crustal cooling during the Phanerozoic, with the density of ZHe age data increasing through the late Miocene. This dataset of thermal information provides a unique opportunity to integrate and analyze cyclicity of magmatic and crustal cooling events for much of the north central and northern Andes.

## Discussion

### Provenance of Amazon Fan sediment

The distribution of DZ U-Pb and U-Th/He double dates from sediment of the Amazon Fan is representative of clastic detritus transferred from the Amazon River to the Atlantic Ocean over late Pleistocene timescales (Figs. 1 and 2). All U-Pb crystallization age groups from the fan have some component of recent cooling ages, suggesting tectonic exhumation in the Andes (Fig. 3). New DZ double dates support previous interpretations; the majority of the Amazon Fan DZs are derived

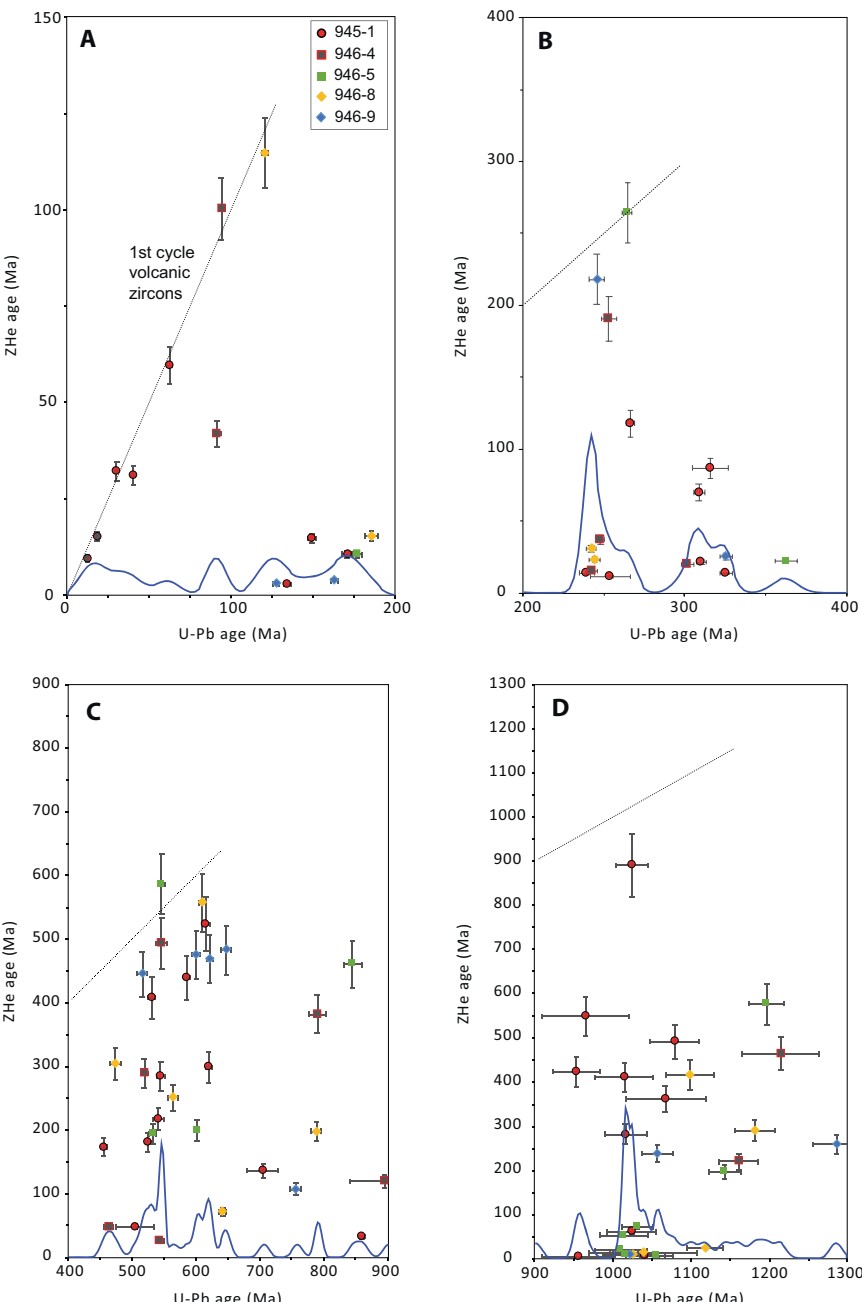

**Fig. 3 | Cross plots for detrital zircon U-Pb and U-Th/He double dates, organized by U-Pb age group. A** 0–200 Ma, **B**: 200–400 Ma, **C**: 400–900 Ma, and **D**: 900–1300 Ma. Kernel density estimates (blue curves) represent the U-Pb age density for each group. Note scale changes across (**A**–**C**). All error bars are two standard deviations.

from the central and northern Andes (Mason et al.[7]), rather than the more areally extensive but low relief Amazon craton. Indeed, while the Andes represent a relatively small fraction (areally) of the Amazon drainage basin (~12%), the Cordilleran system contributes a disproportionate amount of DZs with Cenozoic ZHe cooling ages (38% of total) to the Amazon Fan. DZs with U-Pb ages <400 Ma are likely sourced entirely from the Andes (Fig. 3A, B), as most DZs in this U-Pb age group are either first cycle volcanic zircons, have Triassic – Jurassic ZHe ages, or have Late Cretaceous through Neogene ZHe ages. Although DZs with Andean arc U-Pb ages (*sensu stricto*; 0–50 Ma) are scarce (*n* = 4/112), ZHe cooling ages display the most pronounced mode between 0 and 50 Ma (Fig. 2), demonstrating the importance of tectonic driven erosion of the Cordillera to the ultimate transfer of sediment to the Atlantic margin. This interpretation is consistent with

relatively high rates of denudation measured using [10]Be in modern Andean sourced streams (Wittmann et al.[31]).

Craton aged DZs (>1300 Ma) correspond largely to Neoproterozoic and Paleozoic ZHe cooling ages, which could be interpreted as sediment recycling in the cordillera or primary cratonic sources. Exhumation between 400 and 500 Ma involved uplift and erosion of Sunsás (900–1300 Ma) through early Paleozoic aged rocks, as shown by the U-Pb and ZHe relationships of DZs (Figs. 2 and 3C, D). The mode of ZHe cooling ages between ~400 and 500 Ma are consistent with a documented long-lived, and spatially continuous active margin (the Famatinian orogeny) along western Gondwana that was roughly contemporaneous with the Taconic orogeny of Laurentia (Chew et al.[36]). Although Neoproterozoic through craton aged DZs are documented in Andean rivers (Capaldi et al.[27]; Jackson et al.[22]), Cenozoic cooling ages

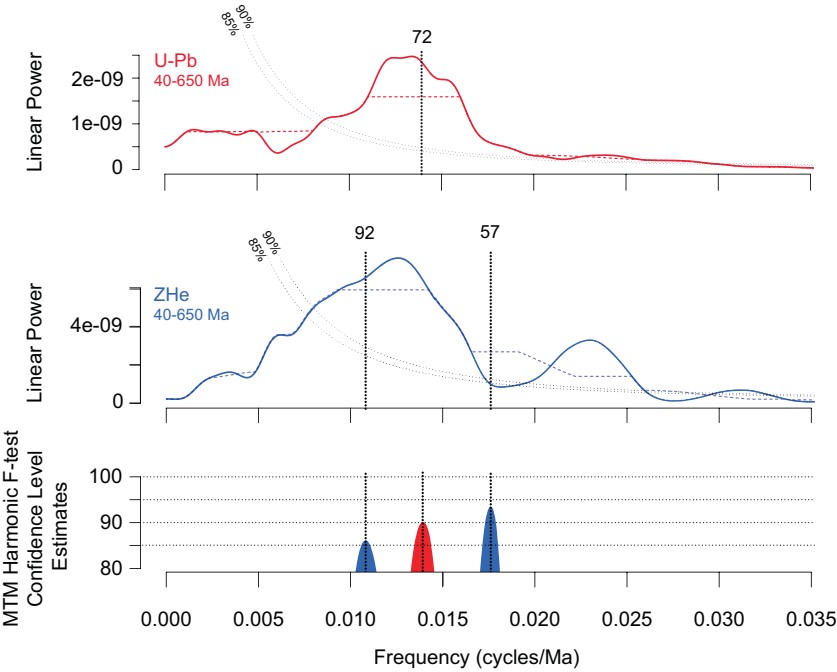

**Fig. 4 | Spectral characteristics of the KDE interval between 650 and 40 Ma for the U-Pb and U-Th/He (ZHe) record.** Blue or red dashed lines indicate the median smoothed spectrum (Meyers[52]). Peaks are only identified if they satisfy both the ML96 robust red noise model (Mann and Lees[53]) and MTM harmonic *F*-test at 85% and 90% confidence levels (e.g., Meyers[54]). Here, we identify two periods of 92 Ma and 57 Ma in the ZHe record, and a periodicity of 72 Ma in the DZ U-Pb record from the Amazon Fan.

associated with those U-Pb age groups provides another line of evidence that recycling (sedimentation, burial heating, and later exhumation) of older cratonic DZs is common in the Cordillera of South America.

### Past cycles of orogenesis preserved in the Amazon Fan

Our frequency analyses of DZ U-Pb and ZHe age data (Fig. 4) suggest periodicity in magmatic (72 Myr cycles) and crustal cooling events (57 and 92 Myr cycles) archived in the Amazon Fan, which we interpret to be related to the long history of upper and lower plate coupling along the subduction dominated western margin of South America. The conceptual model suggested by DeCelles et al.[2] explicitly predicts an association between high magmatic flux events and creation of orogenic relief, driven by underthrusting of melt fertile crust, and isostatic surface uplift in response to delamination of dense lithospheric mantle beneath the orogen. Before or during such a phase of high magmatic flux, the width of the orogen is thought to grow as the fold-thrust belt propagates, resulting in exhumation and cooling of crustal material. In our dataset, distinct modes of ZHe cooling ages partially overlap with, precede, or temporally lag modes of U-Pb ages, suggesting a relationship between magmatic flux events (modes of U-Pb ages), and enhanced crustal cooling (ZHe age modes; Fig. 5A, B) in the north central and northern Andes. Carrapa and Decelles[4] interpreted an anticorrelation between magmatic flux, deformation of the orogen, and crustal cooling, and suggested deformation temporally precedes magmatic flux. Our dataset is consistent with models linking magmatism and deformation (Fig. 5), though our data do not discriminate between competing models for mechanisms of cyclic orogenesis.

The periods of orogenic activity, as archived in the Amazon fan, are reasonably consistent with prior estimates of cyclicity in the central to northern Andes (25–50 Myr; DeCelles et al.[2]; ~70 Myr; Sundell et al[5]; Sundell et al.[17]). Previous reports of timescales for cyclicity were focused on relatively specific and restricted geographic extents during the Mesozoic through Cenozoic. The Amazon Fan samples both the

north central and northern Andes, with a ZHe record that spans the Phanerozoic. This broad geographic and temporal sampling may explain moderate differences in reported periodicity across multiple studies, especially given the more recent record of diachronous mountain building events along strike in the central to northern Andes (e.g., Boschman[36]).

Figure 6 examines the recent (0–200 Ma) relationship between magmatic flux events and crustal cooling recorded by the Amazon Fan, and further highlights an apparent temporal relationship between magmatic events and increased crustal cooling. Here, a Paleogene increase in ZHe age density is followed by a minor lull at ~40 Ma, and a subsequent major Neogene increase; a pattern which coincides broadly with geographically broad temporal patterns of coarse-grained sedimentation, a reliable signal of uplift in the hinterland (Painter et al.[37]), to the Andean retroarc foreland from Columbia, Central Peru, and southern Bolivia (Horton[38]; Horton[11]).

Here, results of detrital geo- and thermochronology (Fig. 6) are consistent with cyclic orogenic phenomena proposed for Andean style mountain building (DeCelles et al.[3]; DeCelles et al.[2]; Sundell et al.[5]). For example, Jurassic through Neogene orogenesis is well documented during the formation of the modern Andes (Garzione et al.[16]; Horton[11]); ZHe ages from the Amazon Fan record phases of magmatism and increased crustal cooling during the Jurassic (~180 Ma), Cretaceous (~110 Ma) and latest Cretaceous – early Paleogene (~60–70 Ma), and early late Miocene (~12 Ma) (Fig. 6) that track with known orogenesis in the Andes.

We hypothesize numerous exhumational cooling events remain recorded in un-reset ZHe cooling ages from upland Andean bedrock areas of the eastern Cordillera (e.g., Reiners et al.[39]), for example in the extensive Paleozoic metamorphic and siliciclastic units exposed in the eastern Andes of Bolivia and Peru (Gomez Tapias et al.[23]). This would explain the abundance of Paleozoic ZHe cooling ages found in what is hypothesized to be largely Andean derived clastic sediment from the northern central and northern Andes.

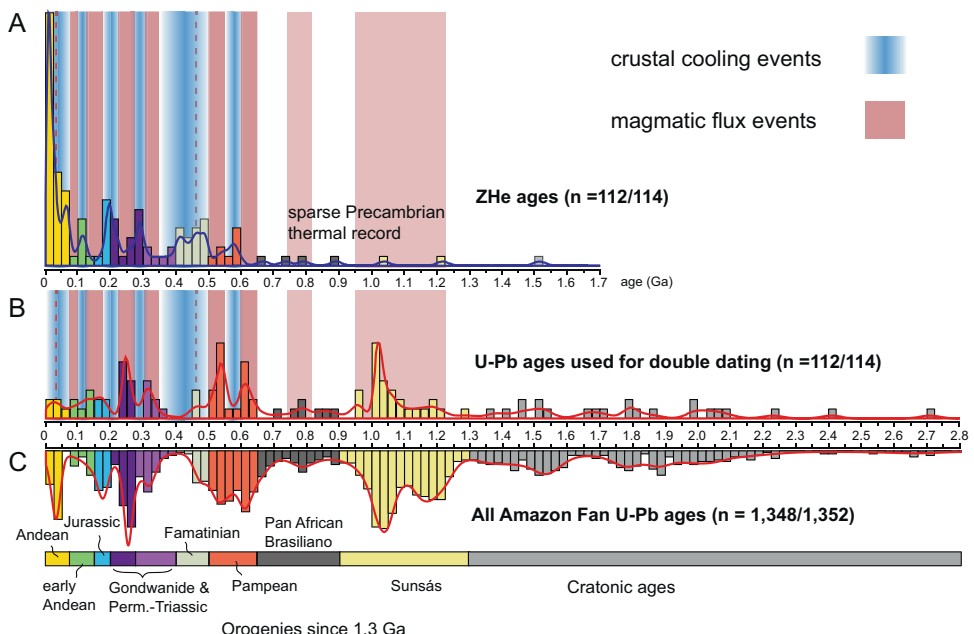

**Fig. 5 | Kernel density estimates (KDEs) for detrital zircon U-Th/He age data and U-Pb age data from the Pleistocene Amazon Fan, highlighting temporal relationships between South American tectonic events, periods of magmatic flux (colored red), and pulses of exhumation-driven cooling (colored blue).** **A** KDE and histogram (25 Ma bins) for U-Th/He age data. **B** KDE and histogram for U-Pb age data associated with cooling ages in part (**A**). **C** KDE and histogram for all published U-Pb DZ age data from the Amazon Fan (Mason et al.[7]).

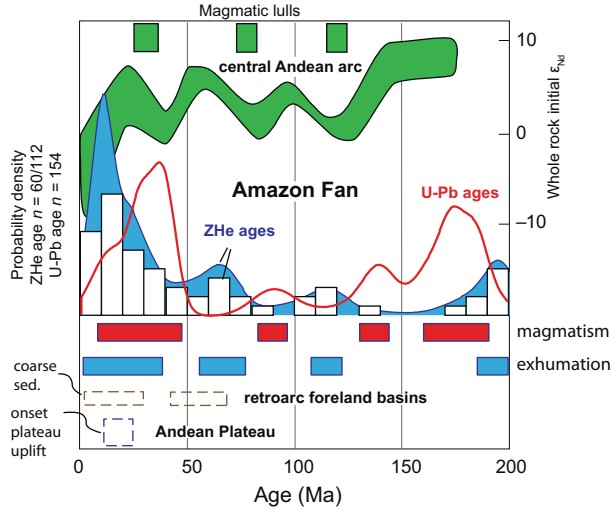

**Fig. 6 | A 200 Ma record of magmatism and crustal cooling for the north central to northern Andes inferred from detrital zircon U-Pb and U-Th/He from the Amazon Fan, including continental records of the timing of coarse-grained clastic sedimentation to Andean retroarc foreland basins (Horton[38]; see main text for discussion), and initiation of widespread Andean plateau uplift (Sundell et al.[17]).** Top green curve shows whole rock neodymium values used to interpret central Andean arc magmatic magmatism, including lulls, since ~175 Ma (modified from DeCelles et al.[3]). Unfilled red curve is the KDE of U-Pb detrital zircon (DZ) ages from the Amazon Fan (Mason et al.[7]) with interpreted phases of high magmatic flux since 200 Ma. Blue filled curve and histogram are DZ U-Th/He cooling ages (AHe) from the Amazon Fan, with interpreted phases of rapid or widespread crustal cooling.

Here, we show ZHe cooling ages from the Amazon Fan—a reliable repository of the entire continent-scale catchment—can be viewed as a detrital record of as many as six distinct phases of exhumation-driven cooling along western Gondwana – western South American margin since the late Neoproterozoic (Figs. 5, 6).

## Methods

### Double dating (U-Pb and U-Th/He) of Amazon Fan detrital zircons

DZ U-Pb ages are rarely reset during multiple phases of orogenesis, hindering interpretations of geologically recent and shallow exhumation histories (<5 km). Although quantitative mixture models applied to DZ age spectra have improved sediment provenance interpretations in the Amazon and other systems (Mason et al.[40]; Fildani et al.[41]; Mason et al.[7]; Blum et al.[42]), double dating of DZs significantly improves the interpretations of provenance, and more so, the geologically recent thermotectonic histories experienced by DZs (Reiners[43,44]; Campbell et al.[34]; Fildani et al.[8]; Thomson et al.[43]; Odlum et al.[45]; Odoh et al.[46]; Fosdick et al.[47]).

ZHe thermochronology exploits the thermal diffusivity of $^4$He in zircon produced by radiogenic decay of $^{238}$U, $^{235}$U and $^{232}$Th. At high temperatures (>~200 °C), zircons do not retain measurable amounts of $^4$He; once a zircon is advected through its closure temperature range (or the partial retention zone; 140°–200 °C), the zircon will begin to retain $^4$He in measurable amounts that are a function of He diffusion kinetics and time since passage through the closure temperature. While U-Pb ages record primary crystallization from magmatic sources, U-Th/He dating of individual zircons yields information about the most recent exhumation history experienced by a zircon.

We measured 114 ZHe cooling ages in previously dated (U-Pb) DZs from the Amazon Fan (Fig. 1B). Selection of DZs for double dating was based on U-Pb age criteria; importantly, we sought to produce a representative characterization of cooling histories of detritus from the Amazon basin by measuring ZHe cooling ages in similar proportions to previously defined U-Pb age distributions from the Pleistocene Amazon Fan (see Fig. 1B) (Mason et al.[7]). DZs for double dating were further screened based on grain size, and morphology suitable for the ZHe technique (Wolfe and Stockli[48]; Hart et al.[49]). DZs were selected from five samples of turbiditic sands collected from two sediment cores recovered during Ocean Drilling Program Leg 155 (Sites 945 and 946; see Supplementary Fig. S1), which are proximal to one another and easily correlated (see Flood et al.[50]; Mason et al.[7]). We amalgamate

these subsamples (See Supplemental Fig. S1) to better characterize complex thermotectonic histories of sediment from the Amazon Fan. This approach integrates sedimentation on the Amazon Fan over multi-millennial timescales (MIS-6 through MIS-2; ~180–20 ka; Flood et al.[50]).

### Frequency analyses of U-Pb and ZHe age data

In order to quantitatively evaluate the periodicity of hypothesized cyclic orogenic processes (e.g., DeCelles et al.[2]; Sundell et al.[5]) we analyzed the KDE interval between 650 and 40 Ma of the U-Pb and ZHe record (Fig. 4). By excluding the most recent 40 Ma of the record we aimed to avoid distortion of the spectrum, given the significant increase in the data over that portion of the record (Weedon[51]). To estimate the spectral characteristics of the record we used the R package *Astrochron* (Meyers[52]) to carry out power spectra analysis using the Mann and Lees[53] robust red noise Multi-Taper method (MTM) analysis to test for statistical significance of spectral peaks. Prior to analysis the KDE was interpolated to even sample spacing, prewritten using an autoregressive-1 (AR1) filter, and detrended. We report periodicities that exceed the 85% and 90% MTM harmonic *F*-test confidence level estimate, while also achieving the required robust red noise confidence level (85%) within ± half the power spectrum bandwidth resolution. This method of estimation substantially reduces the identification of false positive estimates of spectra peaks (Meyers[54]). The results of the frequency analyses (Fig. 4) show statistically significant periodicity in U-Pb ages that occur at 72 Myr (90% confidence), and ZHe cooling age frequency that occurs at 57 and 92 Myr periods (>90% and 85% confidence level, respectively).

## Data availability

The geochemical datasets generated in this study are provided in Supplementary Data 1.

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

## Acknowledgements

The authors wish to acknowledge Equinor for grant funding.

## Author contributions

C.M., B.W.R., A.F., and D.F.S. contributed to the design of the double dating study. M.O.P. and C.M. coordinated and performed the frequency analyses of U-Pb and ZHe age data. All authors contributed to data interpretation and writing of the paper.

## Competing interests

The authors declare no competing interests.
