## [Peer Review File · Nature Communications]

Cycles of Andean mountain building archived in the Amazon fanREVIEWER COMMENTS

Reviewer #1 (Remarks to the Author):

General remarks:

This manuscript presents the double dating of detrital zircons (U-Pb crystallization ages and U-Th/He cooling ages) collected from the late Pleistocene Amazon Fan sediments, and aims to illustrate the thermotectonic evolution history and sediment transfer patterns in the Amazon drainage basin. The major finding of this work is the recognition of periodicity in thermal events in the central and northern Andes, i.e. ~ 57 and ~ 92 Myr recorded in the ZHe cooling ages, and the authors further interpreted this periodicity in thermal events could be regarded as the thermotectonic signals of recurring geodynamic processes that resulted in high fluxes of magma to the arc in the western margin of South America. Overall, the use of double dating method can provide more robust constraints on the sources and origins of detrital zircons in this large and complex drainage basin. For instance, the authors found the evidences for recycling and recent exhumation of craton aged detrital zircons. The major findings of this work may attract the research attentions of the communities working on orogenic, thermotectonic and sedimentary basin evolutions in continental margins. Thus, the major topic of this paper may fit the scope of this international prime journal although it may be more suitable for some professional journals such as GSAB, Basin Research, and Tectonophysics.

My major concerns on this paper lie in the data quality and interpretation:

- 1) Detrital zircon fertility has been widely documented in recent years for it may greatly influence the source discrimination and recognition of zircon ages in the detrital sediments. As the manuscript shown here, the Amazon basin is featured by diverse source rocks and complex provenance lithology associated with the multi-phase thermotectonic evolution. The influence of detrital zircon fertility in the Andean upland area on the zircon source-to-sink transport process in the Amazon Basin has to be considered first.
- 2) Sedimentary evolution of detrital zircon ages. The detrital zircons were separated from the late Pleistocene Amazon Fan sediments spanning from MIS 6 to MIS 2. Considering the possible difference in sediment erosion and transport patterns in the large basin in response to climatic evolution, I wonder whether there is temporal changes in detrital zircon ages during glacial and interglacial periods in the late Pleistocene.
- 3) Periodicity in thermal events. It is interesting to get these two cyclicity of ~ 57 and ~ 92 Myr. Nevertheless, this manuscript did not discuss the geological meaning of the periodicity. Do they correspond to some mechanism of plate subduction and/or crustal growth processes? Is there any analog in the other places?

More specific comments:

1. L103-105: Supportive references?
2. L148-149: What exact meaning of these recycled cratonic zircons? Do you mean these zircons underwent multiple cycles in the craton evolution or they were ultimately derived from recycled craton?
3. L158-161: It's interesting to note the provenance results from coarse-grained detrital zircons are almost the same with those fine-grained sediments. They may have much different sediment routing processes especially in the large river basins.
4. L183-184: the grain size of detrital zircons for dating?
5. L233-234: Please note that the residence time of detrital zircons is totally different from the bulk sediment in a specific river system.
6. L311, Section 5.1: See comment above. Thus, the zircon ages only reflect the provenance of detrital zircons, not the bulk Amazon sediments.
7. L360-361: Longer than previous estimates. Why?
8. L382-383: More lines of evidences for this hypothesis?
9. L401: How about the temporal variations of these zircon double ages during the late Pleistocene?
10. L411: Geological meanings of this periodicity?

Reviewer #2 (Remarks to the Author):

Dear editor and authors,

I have now completed the revision of the manuscript entitled "Cycles of Andean Mountain Building Archived in the Amazon Fan" by Mason and colleagues. This is a very interesting project with new thermochronological data in detrital zircons of the Amazon Fan. The findings and interpretations presented in the manuscript are consistent with orogenic cycles in the western margin of South America.

I recommend minor modifications, which are highlighted in the attached author's PDF files. However, I suggest one particular modification for clarity purposes. In Section 4.5, the authors indicate temporal gaps in cooling of crustal material in the presented dataset. How is it possible to identify Famatinian-Pampean cooling ages (600-400 Ma) and not at 250-225 Ma? Such period corresponds to cooling of a tectonically overthickened crust (Pangea amalgamation) and the onset of extensional setting, leading to Pangea breakup during the Late Triassic–Early Jurassic. Igneous and metamorphic rocks associated with this thermal event are exposed in Argentina, Chile, Ecuador, Peru, Bolivia, Colombia, and Venezuela (see Spikings et al., 2016, Gondwana Res). I suggest expanding on this discussion.

I hope this review is helpful, and I am looking forward to seeing the manuscript published.

Best regards,
M. Daniela Tazzo-Rangel

Response to Reviewer Document. December 2021

Dear Reviewers,

This document contains responses to Reviewer comments for the manuscript titled “Cycles of Andean Mountain Building Archived in the Amazon Fan” (NCOMMS-21-37985-T). Our responses are written in red, and reviewer comments are written in black.

We pasted new text from the revised manuscript directly into this document where appropriate, as a response to many suggested revisions.

Reviewer #1 (Remarks to the Author):

General remarks:

This manuscript presents the double dating of detrital zircons (U-Pb crystallization ages and U-Th/He cooling ages) collected from the late Pleistocene Amazon Fan sediments, and aims to illustrate the thermotectonic evolution history and sediment transfer patterns in the Amazon drainage basin. The major finding of this work is the recognition of periodicity in thermal events in the central and northern Andes, i.e. ~57 and ~92 Myr recorded in the ZHe cooling ages, and the authors further interpreted this periodicity in thermal events could be regarded as the thermotectonic signals of recurring geodynamic processes that resulted in high fluxes of magma to the arc in the western

margin of South America. Overall, the use of double dating method can provide more robust constraints on the sources and origins of detrital zircons in this large and complex drainage basin. For instance, the authors found the evidences for recycling and recent exhumation of craton aged detrital zircons. The major findings of this work may attract the research attentions of the communities working on orogenic, thermotectonic and sedimentary basin evolutions in continental margins. Thus, the major topic of this paper may fit the scope of this international prime journal although it may be more suitable for some professional journals such as GSAB, Basin Research, and Tectonophysics.

We thank Reviewer #1 for the succinct summary of this work. We feel the dataset of detrital zircon double dates will be of interest across geological fields, as you noted above.

One point of clarification, the Reviewer states that our major finding is “...*the recognition of periodicity in thermal events in the central and northern Andes, ...*”

While this recognition is important, we hope that Readers, Reviewers, and Editors will not overlook that a first-order finding is the high proportion of young (Andean) cooling ages from detrital zircons spanning a huge range of U-Pb (crystallization) ages. This point is the first topic of our *Discussion*.

My major concerns on this paper lie in the data quality and interpretation:

We have focused on improving and clarifying our interpretations.

1) Detrital zircon fertility has been widely documented in recent years for it may greatly

influence the source discrimination and recognition of zircon ages in the detrital sediments. As the manuscript shown here, the Amazon basin is featured by diverse source rocks and complex provenance lithology associated with the multi-phase thermotectonic evolution. The influence of detrital zircon fertility in the Andean upland area on the zircon source-to-sink transport process in the Amazon Basin has to be considered first.

Thank you for this thoughtful comment.

As a response, we added the following statement and reference to the text of our revised manuscript:

“Differential zircon fertility of source terranes has been shown to influence DZ age distributions in some cases (Moecher and Samson, 2006), which may interfere with provenance interpretations. However, denudation rates of Andean streams that are an order of magnitude higher than low-lying Amazonian streams are likely a dominant process driving provenance signatures in the Amazon Fan. Given the source of sediment in the Amazon Fan is largely from the cordilleran system, the Amazon Fan sedimentary record should preserve the long thermal history of the tectonically active western margin of South America.”

To expand on why we chose to limit our discussions on zircon fertility:

(1) when it comes to Andean vs craton sources of DZs, we think denudation rate is the principle control on DZ age distributions found in the Amazon River and fan. There are some supporting lines of evidence, including measurements of cosmogenic radionuclides in modern streams (very high erosion in Andean streams vs very low in craton) and direct measurements of modern sediment concentrations/loads that show

Andean tributaries have much higher denudation rates (~**10x**) and sediment loads than those of low-lying tributaries to the Amazon River/Fan.

(2) Unfortunately, no existing datasets (that we are aware of) document zircon fertility across major streams draining the Andes, thus our dataset does not allow us to discriminate effects (if measurable) of differential fertility over the spatial scales of the Andes or wider Amazon River basin.

In summary zircon fertility is not likely to be an issue (importance of relative erosion rates in Andes vs low-lying areas), and not necessarily possible to address here (lack of fertility datasets).

2) Sedimentary evolution of detrital zircon ages. The detrital zircons were separated from the late Pleistocene Amazon Fan sediments spanning from MIS 6 to MIS 2. Considering the possible difference in sediment erosion and transport patterns in the large basin in response to climatic evolution, I wonder whether there is temporal changes in detrital zircon ages during glacial and interglacial periods in the late Pleistocene.

This is an interesting idea, one which we investigated and addressed using U-Pb age data in Mason et al., (*GEOLOGY*; 2019). **In short, detrital zircons from the fan appear to be well mixed.** In our 2019 paper, we specifically wrote: “*KDE cross-correlation coefficients for all Amazon Fan samples (see Table DR2 in the Data Repository) have a mean and standard deviation (1σ) of 0.45 ± 0.08 , which is similar to or slightly less than coefficients for synthetic subsamples drawn from the same parent sample (Saylor and Sundell, 2016).*”

Our statistical analysis of 10 DZ samples led us to conclude that age spectra, and thus *potentially* sediment source area, did not vary significantly over 100 kyr timescales, and that any observed variability in age spectra between individual samples could be due to incomplete sampling of the same age distribution. Although our new cooling ages do show significant variability across the five subsamples, **this can almost certainly be attributed to incomplete sampling**, or relatively low numbers of ages (n=14 to n=46 individual ages per subsample). It would be difficult to draw robust interpretations solely from individual subsample age distributions of 15-20 zircons.

Logically, if the available evidence suggests that the DZ age mixture isn't changing much through time (over 100 kyr timescales, as in Mason et al., 2019), then we argue that amalgamating is the best method to draw more robust conclusions from the higher-n dataset. We feel that published interpretations support our approach of amalgamating five subsamples here.

3) Periodicity in thermal events. It is interesting to get these two cyclicity of ~57 and ~92 Myr. Nevertheless, this manuscript did not discuss the geological meaning of the periodicity. Do they correspond to some mechanism of plate subduction and/or crustal growth processes? Is there any analog in the other places?

This is a natural follow up question in light of the interpretations of our dataset. We included a new analysis of the DZ U-Pb record, which results in similar timescales of cyclicity as the ZHe record, suggesting links between magmatism and exhumation/cooling. We do offer some discussion in the revised manuscript: "Our frequency analyses of DZ U-Pb and ZHe age data suggest periodicity in magmatic (72 Myr

cycles) and crustal cooling events (57 and 92Myr periods) archived in the Amazon Fan, which we interpret to be related to the long history of upper and lower plate coupling along the subduction dominated western margin of South America. The conceptual model suggested by DeCelles et al. (2015) explicitly predicts an association between high magmatic flux events and creation of orogenic relief, driven by underthrusting of melt fertile crust, and isostatic surface uplift in response to delamination of dense lithospheric mantle beneath the orogen. Before or during such a phase of high magmatic flux, the width of the orogen is thought to grow as the fold-thrust belt propagates, resulting in exhumation and cooling of crustal material. In our dataset, distinct modes of ZHe cooling ages partially overlap with, precede, or temporally lag modes of U-Pb ages, suggesting a relationship between magmatic flux events (modes of U-Pb ages), and enhanced crustal cooling (ZHe age modes; Fig. 5 A, B) in the north central and northern Andes. Carrapa and Decelles (2015) interpreted an anticorrelation between magmatic flux, deformation of the orogen, and crustal cooling, and suggested deformation temporally precedes magmatic flux. Our dataset is consistent with models linking magmatism and deformation (Fig. 5), though our data do not discriminate between competing models for mechanisms of cyclic orogenesis.”

As you can see, we reference several well-cited publications dealing with the proposed mechanisms and timescales of cycles of orogenesis, and we DO discuss our results in context of these existing studies.

More specific comments:

1. L103-105: Supportive references?

We added a reference to Horton (2018a) here.

2. L148-149: What exact meaning of these recycled cratonic zircons? Do you mean these zircons underwent multiple cycles in the craton evolution or they were ultimately derived from recycled craton?

Clarified in the revised ms.

But FYI we mean recycled in a sedimentary basin within the Andes, probably. In other words, they were eroded from the craton, incorporated into the Cordillera where perhaps buried deeply and reset (or not?), and later eroded again during uplift and erosion, and transferred to the Amazon fan (recycled).

3. L158-161: It's interesting to note the provenance results from coarse-grained detrital zircons are almost the same with those fine-grained sediments. They may have much different sediment routing processes especially in the large river basins.

It is interesting, and we are currently looking at provenance of fine-grained deposits in the fan... But here we aren't making any inferences about *sediment residence or travel times*, just noting that several studies agree, the Andes provide the majority of sediment, apparently across grain sizes.

4. L183-184: the grain size of detrital zircons for dating?

These are five subsamples from Mason et al 2019—dominantly fine grained turbidite sands.

5. L233-234: Please note that the residence time of detrital zircons is totally different from the bulk sediment in a specific river system.

Fair statement. And again, we don't make any inference about residence times here.

6. L311, Section 5.1: See comment above. Thus, the zircon ages only reflect the provenance of detrital zircons, not the bulk Amazon sediments.

Thank you.

7. L360-361: Longer than previous estimates. Why?

Interesting question.

We added the following text to the revised version:

“Previous reports of timescales for cyclicity were focused on relatively restricted geographic extents, while the Amazon Fan samples both the Northern and North Central Andes. This broad geographic sampling may explain differences in reported periodicity across multiple studies, especially if diachronous mountain building along strike in the central to northern Andes influences the resulting spectra of ZHe ages in the Amazon Fan (e.g. Boschman, 2021; ESR).”

8. L382-383: More lines of evidences for this hypothesis?

We added the following text and reference: “We hypothesize numerous exhumational cooling events remain recorded in un-reset ZHe cooling ages from upland Andean bedrock areas of the eastern Cordillera (e.g. Reiners et al., 2015), for example in the extensive Paleozoic

metamorphic and siliciclastic units exposed in the eastern Andes of Bolivia and Peru (Gomez Tapias et al., 2019).

9. L401: How about the temporal variations of these zircon double ages during the late Pleistocene?

I think we address this in point # 2 above.

We don't believe there's much variation (Mason et al 2019), and it is not possible to draw meaningful conclusions from very low-n samples, thus we amalgamate them to achieve a higher-n and make robust interpretations.

10. L411: Geological meanings of this periodicity?

I think we address this in point #3 above.

Reviewer #2 (Remarks to the Author):

Dear editor and authors,

I have now completed the revision of the manuscript entitled "Cycles of Andean Mountain Building Archived in the Amazon Fan" by Mason and colleagues. This is a very interesting project with new thermochronological data in detrital zircons of the Amazon Fan. The findings and interpretations presented in the manuscript are consistent with orogenic cycles in the western margin of South America.

I recommend minor modifications, which are highlighted in the attached author's PDF files. However, I suggest one particular modification for clarity purposes. In Section 4.5, the authors indicate temporal gaps in cooling of crustal material in the presented dataset. How is it possible to identify Famatinian-Pampean cooling ages (600-400 Ma) and not at 250-225 Ma? Such period corresponds to cooling of a tectonically overthickened crust (Pangea amalgamation) and the onset of extensional setting, leading to Pangea breakup during the Late Triassic–Early Jurassic. Igneous and metamorphic rocks associated with this thermal event are exposed in Argentina, Chile, Ecuador, Peru, Bolivia, Colombia, and Venezuela (see Spikings et al., 2016, Gondwana Res). I suggest expanding on this discussion.

Thank you for the comment and suggestion. Because we did not return to the concept of lulls in the Discussion section, we decided to cut this portion of text from the revised manuscript.

However, it is an interesting feature of our data, the apparent period of decreased crustal cooling (225 – 250 Ma) is, as you noted, interpreted only through a lens of DZ AHe cooling ages, and not through the DZ U-Pb ages. In fact, that period of time is bookended by large increases in crustal cooling 200 – 225, and 275 – 300 Ma. Also, *the DZ U-Pb record does show a major mode between 225 – 275 Ma*, suggesting that a magmatic/volcanic/metamorphic history from this period of time is well preserved in the Amazon fan.

I hope this review is helpful, and I am looking forward to seeing the manuscript published.

Thank you!

We made the text in Fig 1. C more legible.

REVIEWERS' COMMENTS

Reviewer #1 (Remarks to the Author):

General remarks:

This revised manuscript made a great improvement by incorporating the major comments and suggestions raised by the reviewers. The authors have well addressed my concerns on the influence of detrital zircon fertility on the zircon source-to-sink transport process in the Amazon Basin although they do not provide or find existing data on the spatial distribution of detrital zircon ages.

As the reply said, quite a lot of previous studies have reported the high denudation rates in high Andean watersheds and the low rates in lowland plains. The latest data on meteoric Be-10 by Hella Wittman et al. may be included in figure 1.

As for the temporal changes in detrital zircon ages in the late Pleistocene, they argue that the detrital zircons from the fan appear to be well mixed and the literature data support their amalgamating method. I still suggest more sample analyses will definitely provide more robust constraints and better support your conclusion.

I have no major concerns on this revised version and it may be accepted for the publication after some refinement.

Some detailed comments and suggestions:

1. Figure 1: the lithostratigraphy in 1C is too small to be readable. Better change "erosion rate" to "denudation rate" in the legend of 1A.
2. Figure 6: it is not very clear and needs some refinement. The "coarse-grained sedimentation" indicates sandy sediment or gravel?
3. Datasets: The meaning or indications of some colored or shaded numbers in the excel sheets need to be given. Some sheets look messy.

Response to Reviewer Document. October 1, 2022

Reviewer #1 (Remarks to the Author):

General remarks:

This revised manuscript made a great improvement by incorporating the major comments and suggestions raised by the reviewers. The authors have well addressed my concerns on the influence of detrital zircon fertility on the zircon source-to-sink transport process in the Amazon Basin although they do not provide or find existing data on the spatial distribution of detrital zircon ages.

Thank you for the prior comment RE to improve the discussion of uncertainty with respect to zircon fertility. To respond to the above comment, and just to clarify, in the last response to reviewer document we cited a lack of detrital zircon *fertility* estimates at the Amazon watershed scale. This is distinctly different from existing published compilations of DZ *U-Pb* ages at the Amazon watershed scale, which have been summarized in existing literature (Mason et al., 2019; Pepper et al., 2016; Mapes, 2009). Particularly, Mason et al. (2019) used mixture models of DZ *U-Pb* ages from the Amazon fan to show the majority of DZs found in the fan are likely derived from the northern central Andes, with lesser proportions from the northern Andes and from the low-lying Craton.

As the reply said, quite a lot of previous studies have reported the high denudation rates

in high Andean watersheds and the low rates in lowland plains. The latest data on meteoric Be-10 by Hella Wittman et al. may be included in figure 1.

Thank you for the newer Wittmann et al. citation. We have included this newer estimate of denudation based on meteoric $10/9\text{Be}$ ratios in the revised version of Figure 1 and in the updated reference list.

As for the temporal changes in detrital zircon ages in the late Pleistocene, they argue that the detrital zircons from the fan appear to be well mixed and the literature data support their amalgamating method.

Thank you for this clarification.

I still suggest more sample analyses will definitely provide more robust constraints and better support your conclusion.

Yes, more analyses (what kind?) may improve our understanding, however at this phase of the project, we see fit to disseminate the current findings to the community. We do plan to follow up with further analyses meant to investigate aspects of the climatic and tectonic histories archived in the Amazon fan.

I have no major concerns on this revised version and it may be accepted for the publication after some refinement.

We sincerely thank you for your constructive reviews and for your time.

Some detailed comments and suggestions:

1. Figure 1: the lithostratigraphy in 1C is too small to be readable. Better change “erosion rate” to “denudation rate” in the legend of 1A.

We have increased the font size and simplified the key of Fig. 1C.

2. Figure 6: it is not very clear and needs some refinement. The “coarse-grained sedimentation” indicates sandy sediment or gravel?

We improved the clarity of Figure 6 and the caption text.

3. Datasets: The meaning or indications of some colored or shaded numbers in the excel sheets need to be given. Some sheets look messy.